

# Vectorized dataset of roadside noise barriers in China using street view imagery

Zhen Qian [1, 2, 3], Min Chen [1, 2, 3, 4], Yue Yang [1, 2, 3], Teng Zhong [1, 2, 3], Fan Zhang [5], Rui Zhu [6], Kai Zhang [1, 2, 3], Zhixin Zhang [7, 1], Zhuo Sun [1, 2, 3], Peilong Ma [1, 2, 3], Guonian Lü [1, 2, 3], Yu Ye [8], Jinyue Yan [9, 10]

[1]Key Laboratory of Virtual Geographic Environment (Ministry of Education of PRC), Nanjing Normal University, Nanjing, 210023, China
[2]State Key Laboratory Cultivation Base of Geographical Environment Evolution, Nanjing, 210023, China
[3]Jiangsu Center for Collaborative Innovation in Geographical Information Resource Development and Application, Nanjing, 210023, China
[4]Jiangsu Provincial Key Laboratory for NSLSCS, School of Mathematical Science, Nanjing Normal University, Nanjing, 210023, China
[5]Senseable City Lab, Massachusetts Institute of Technology, Cambridge, MA 02139, USA
[6]Department of Land Surveying and Geo-Informatics, The Hong Kong Polytechnic University, Kowloon, Hong Kong, China
[7]College of Geography & Marine, Nanjing University, Nanjing, PO Box 2100913, P.R. China
[8]Tongji University, Department of Architecture, College of Architecture and Urban Planning, China
[9]Future Energy Center, Malardalen University, 72123 Vasteras, Sweden
[10]Department of Chemical Engineering, KTH Royal Institute of Technology, Stockholm, 10044, Sweden

*Correspondence to*: Min Chen (chenmin0902@njnu.edu.cn and chenmin0902@163.com)

**Abstract.** Roadside noise barriers (RNBs) are important urban infrastructures to develop a liveable city. However, the absence of accurate and large-scale geospatial data on RNBs has impeded the increasing progress of rational urban planning, sustainable cities, and healthy environments. To address this problem, this study proposes a geospatial artificial intelligence framework to create a vectorized RNB dataset in China using street view imagery. To begin, intensive sampling is performed on the road network of each city based on OpenStreetMap, which is used as the geo-reference to download 5.6 million Baidu Street View (BSV) images. Furthermore, considering the prior geographic knowledge contained in street view images, convolutional neural networks incorporating image context information (IC-CNNs) based on an ensemble learning strategy are developed to detect RNBs from the BSV images. Subsequently, the RNB dataset presented by polylines is generated based on the identified RNB locations, with a total length of 2,227 km in 215 cities. At last, the quality of the RNB dataset is evaluated from two perspectives: first, the detection accuracy; second, the completeness and positional accuracy. Specifically, based on a set of randomly selected samples containing 10,000 BSV images, four quantitative metrics are calculated, with an overall accuracy of 98.61 %, recall of 87.14 %, precision of 76.44 %, and F1-score of 81.44 %. Moreover, a total length of 254 km of roads in different cities are manually surveyed using BSV images to evaluate the mileage deviation and overlap level between the generated and surveyed RNBs. The root-mean-squared error for mileage deviation is 0.08 km, and the intersection over union for overlay level is 88.08 % ± 2.95 %. The evaluation results suggest that the generated RNB dataset is of high quality and can be applied as an accurate and reliable dataset for a variety of large-



scale urban studies. The generated vectorized RNB dataset and the labelled BSV image benchmark dataset are publicly available at https://doi.org/10.11888/Others.tpdc.271914 (Chen, 2021).

## 1 Introduction

In recent years, several studies have documented the substantial impact of traffic noise problems in cities (Apparicio et al., 2016; Begou et al., 2020). Roadside noise barriers (RNBs) are vital urban infrastructure that contribute significantly to
mitigate undesirable traffic noise in communities (Abdulkareem et al., 2021; Ning et al., 2010). Additionally, RNBs contribute to the development of sustainable cities in many ways. For example, with the emphasis on new energy, RNBs are being used to install solar photovoltaic panels, thereby increasing the utility of new energy sources (Gu et al., 2012; Zhong et al., 2021). Besides, the reasonable presence of RNBs enables the airflows in the urban canyon region to be adjusted, thereby improving the roadside air quality (Huang et al., 2021; Zhao et al., 2021). Because of the importance of RNBs in building
sustainable cities, the demand for RNBs has increased alongside traffic growth in recent decades (Boer and Schroten, 2007; Oltean-Dumbrava and Miah, 2016). There are bottom-up benefits from establishing an accurate and standardised large-scale RNB dataset with detailed geospatial information about RNBs, including their mileages, locations, and distributions (Liu et al., 2020; Wang and Wang, 2021). Particularly, precise RNB locations enables traffic departments to effectively manage and maintain this type of infrastructure (Sainju and Jiang, 2020). Additionally, urban research can simulate dynamic cities based
on accurate RNB geospatial information (Wang and Wang, 2021; Zhao et al., 2017). Moreover, governments can rely on the RNB maps to examine urban layouts and create green and sustainable cities (Song et al., 2021; Song and Wu, 2021).
Over the past few years, extensive geospatial databases have been established to store data on many aspects of urban infrastructure (Griffiths and Boehm, 2019; Perkins and Xiang, 2006). However, the sharing and exchange of RNB data in these databases are restricted, and the data only covers a limited geographic area (Wang et al., 2019; Zhang et al., 2022).
These challenges to data acquisition are because databases have to adhere to various standards related to geographic data (e.g., file format and geographic coordination reference) (Lafia et al., 2018). Moreover, the RNB data are often created and updated manually through road inspections and investigations, which are costly and time consuming, especially on a large scale (Potvin et al., 2019; Ranasinghe et al., 2019). Therefore, there is an urgent requirement to seek alternate effective ways to generate and update the RNB geospatial dataset.
Street view imagery is geo-referenced data densely covering the road network of cities. As a new geospatial data source, it provides depictions of real-world surroundings, including natural landscapes and built environment, and enables users to recognize physical objects, urban dynamics features, and geographic scenes on a large scale (Zhang et al., 2018). In addition, as part of the data sharing movement, an increasing number of community-based organizations and corporations, such as Baidu Maps, Tencent Maps and Google Maps, are regularly generating and updating open-access street view imagery (Qin
et al., 2020; Zhang et al., 2019). As a result, such big data brings great prospects for acquiring urban infrastructure information (e.g., RNBs), with benefits such as broad coverage, rapid update speed, and low acquisition cost (Kang et al.,



2020). However, manual interpretation is a tedious task and conventional computer vision algorithms struggle when confronted with large amounts of data and complex image features (Zhang et al., 2018).

With the advancement of computing hardware and frameworks, deep learning methods now have an increased capacity for

extracting semantic features from a large amount of data (Lecun et al., 2015). The emerging approaches are increasingly being used to interpret physical objects and detect interior patterns from earth observation data (Jiang et al., 2021). Meanwhile, image classification based on deep learning has been used to identify RNBs using street view imagery (Zhong et al., 2021). However, for the purposes of identifying RNBs, prior geographic knowledge, which is essential, is frequently overlooked, such as the fact that RNBs are frequently located between roads and densely populated regions (e.g., residential,

educational, and medical areas) (Arenas, 2008; Wang et al., 2018; Zhang et al., 2022). In recent years, a new framework of data-driven research based on geospatial artificial intelligence (GeoAI) and machine learning has resulted in multiple notable improvements in the discovery of geographic scene knowledge (Goodchild and Li, 2021; Li, 2020). When empirical and prior spatial information is included into deep learning approaches, it can help develop a more holistic understanding of a research subject and mitigate the effects of data scarcity or representational bias (Janowicz et al., 2019; Qian et al., 2020). As

a result, it is possible to enhance the effectiveness of deep learning methods for identifying RNBs by incorporating some prior geographic knowledge from street view imagery. Additionally, Wolpert and Macready (1997) introduced the "no free lunch" theory, demonstrating that a single model must pay for some accuracy by degrading its generalizability. This is acceptable, as it is challenging to construct a perfect solution for all scenarios using a single model, particularly when dealing with vast volumes of data and large-scale areas (Wang and Li, 2021).

The purpose of this study is to build an accurate and nationwide vectorized RNB dataset utilizing Baidu Street View (BSV) imagery. To improve the performance of detecting RNBs, this work proposes a GeoAI framework. Concretely, an ensemble of convolutional neural networks incorporating image context information (IC-CNNs) is developed, which considers the prior geographic knowledge contained in street view images. Subsequently, a post-processing method is applied to generate the vectorized RNB dataset based on the identified RNB locations. At last, the RNB dataset quality is quantitatively

evaluated from two perspectives, i.e., the detection accuracy as well as the completeness and positional accuracy. The main contributions of this study can be summarized as follows:

(1) This study provides the first reliable and nationwide vectorized RNB dataset in China, as well as the labelled BSV images which can be used as a benchmark dataset.

(2) A GeoAI framework is presented for processing numerous BSV images in order to generate the RNB mapping and for

comprehensively evaluating the generated results.

(3) This study presents multiple IC-CNNs based on prior geographic knowledge and ensemble learning strategy to achieve high-performance object identification from street view imagery.

The remainder of this paper is organized as follows. Section 2 briefly describes the data and methods used to generate and evaluate the RNB dataset. Section 3 presents the results of the RNB mapping as well as evaluation and analysis for RNB



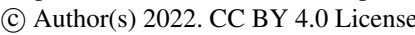

dataset. Section 4 discusses the capability of proposed methods, as well as the challenges and limitations of this work. The last section provides the conclusions of this study.

## 2 Data and methods

### 2.1 The GeoAI framework

The GeoAI framework's workflow is divided into three stages: data preparation, modelling, and evaluation, as shown in Fig.
1. To begin with, BSV images are gathered during the data preparation stage using OpenStreetMap (OSM) road data and the BSV application programming interface (API). Subsequently, BSV images are used to generate various samples for modelling and evaluation. During the modelling stage, deep learning approaches are used to detect RNBs from the BSV imagery. Using the vectorization post-processing method, the identified and scattered RNB locations are subsequently processed into a vectorized dataset. During the evaluation stage, the quality of the created dataset is quantitatively assessed
in two aspects, i.e., the detection accuracy as well as completeness and positional accuracy.

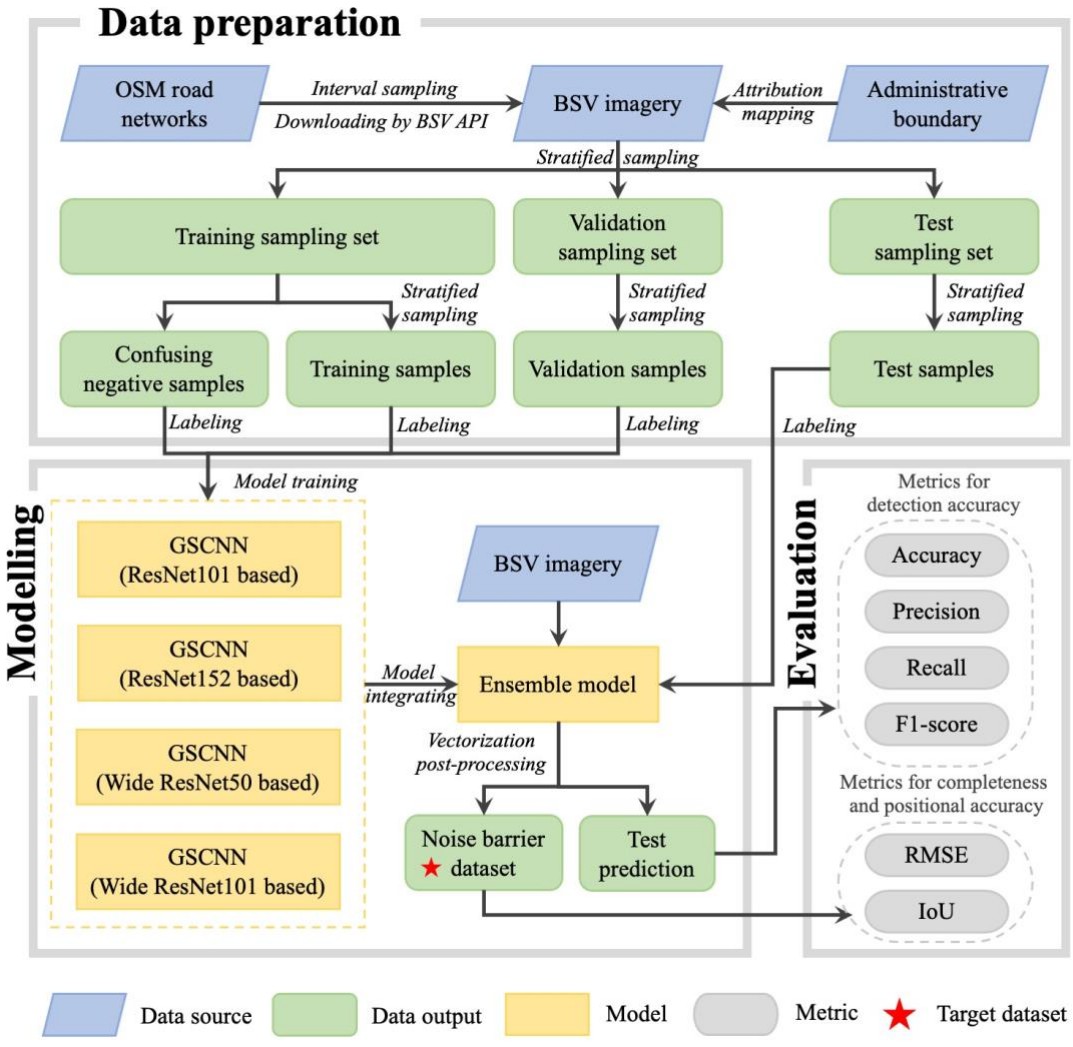

Figure 1: The flow chart of GeoAI framework to generate the vectorized RNB dataset.

## 2.2 Data preparation

Three types of data are acquired for this study: the road networks, administrative boundary, and street view imagery.
Afterwards, training, validation, and test samples are collected based on these data. However, the data from Taiwan province
are scarce.

### 2.2.1 Road networks

The road networks are download from OSM (https://www.openstreetmap.org/) in May 2021, which is polyline-based and
includes a variety of road types, including motorway, trunk road, primary road, and secondary road. According to previous
findings, the quality of OSM road networks in China is high in terms of completeness and positional accuracy (Liu and Long,



2015). In addition, RNBs have a high probability of being installed on motorways and trunk roads (Zhang et al., 2022). Therefore, given the expense of acquiring and computing BSV images, in this study, samples on motorways and trunk roads are only considered for downloading BSV images. Figure 2a depicts the spatial distribution of these two types of roads.

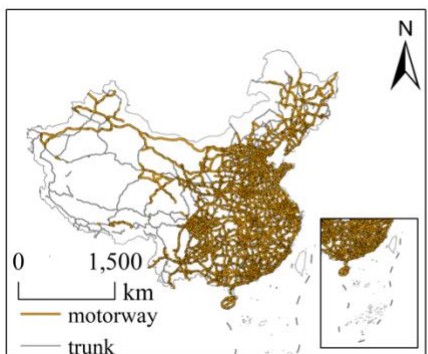
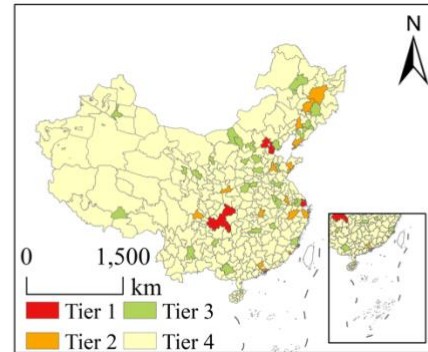
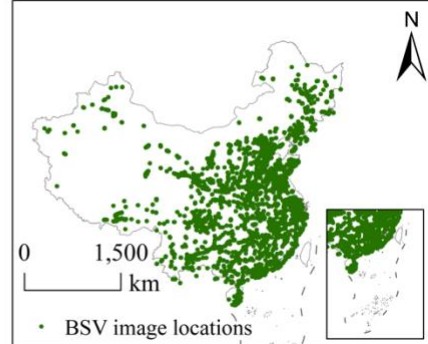

**Figure 2: Three data sources are used in this study. (a) OSM road network data, (b) Chinese administrative boundary with four city tiers, and (c) The locations of the downloaded BSV images. (Road networks are from OSM)**

### 2.2.2 Administrative boundary

The city boundary is acquired from http://bzdt.ch.mnr.gov.cn/ in April 2021. According to the urban management hierarchy established by the Chinese government, cities in China are divided into four tiers (Guan and Rowe, 2018; Jia et al., 2020), including municipalities, sub-provincial cities, prefecture-level cities, and the locations of them are shown in Fig. 2b. Specifically, Tier 1 is centrally administered cities and municipalities. Tier 2 is primarily sub-provincial cities, whereas Tier 3 is province capitals and large prefecture-level cities. Tier 4 is ordinary prefecture cities. Cities with varying administrative levels have varying authorities over resource allocation and jurisdiction (Guan et al., 2018).

### 2.2.3 Street View Imagery

With their high resolution and detailed information on Chinese streets, BSV images are of comparable quality to Google Street View images, which are not available in China (Zhou et al., 2019b). Numerous sample points along OSM roads are collected, and the BSV API is utilized to obtain street view images at those locations. Following the work of Zhang et al. (2022), a sampling interval of around 25 m is utilized to account for the trade-off between data granularity and the expenditure of downloading imagery. As a result, the total number of sample points is 24,871,839. As shown in Fig. 3, the illustration of BSV images with different photography directions shows that BSV image with 90° is more appropriate for the present work because it provides a comprehensive roadside view. Hence, to identify the RNBs along the corresponding roadside, BSV images with a 90° viewing angle are acquired. Owing to the absence of BSV images on a few road segments in a particular year, which will be supplemented in adjacent years. Additionally, the BSV sensors may be obstructed by some vehicles or other surrounding objects. These issues are resolved by downloading multitemporal BSV images from 2014 to



2020. A total of 5,589,691 BSV images are downloaded with a size of 500 pixels × 400 pixels, and their spatial locations are shown in Fig. 2c.

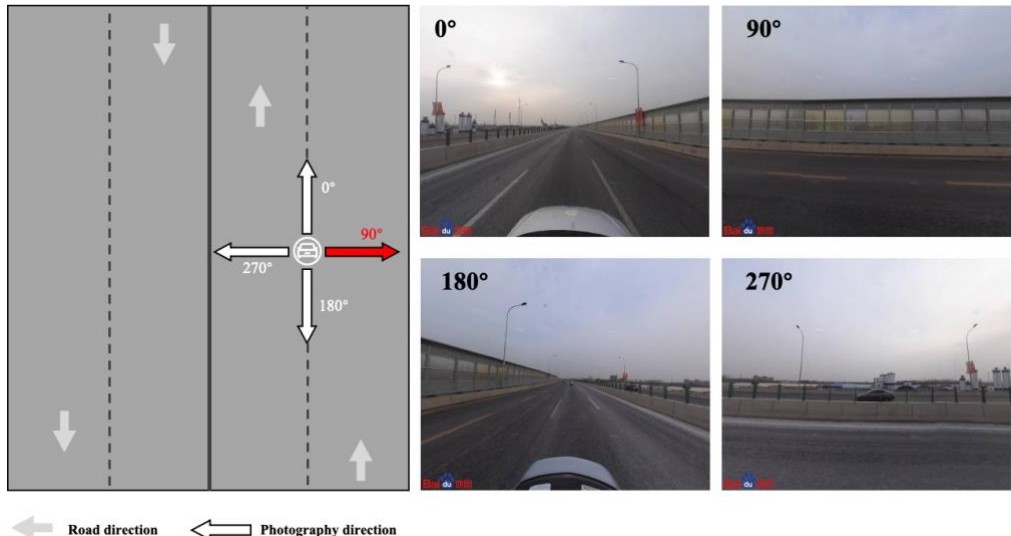

**Figure 3: Illustration of BSV images with different photography directions. (BSV images are from Baidu Maps)**

### 2.2.4 Training, validation, and test sample collection

An effective sampling technique for generating training, validation, and test image samples are developed to detect RNB from the large volume of BSV images collected. Figure 1 illustrates the different steps followed in the data preparation stage. The BSV images are classified into four tiers based on their location within the city administration hierarchy. Subsequently, the training, validation, and test sampling set are subdivided from the entire samples, accounting for 60 %, 20 %, and 20 % of images, respectively. These sampling sets can be used to collect the corresponding samples and benefit by avoiding the

mixing of samples.

Previous investigations revealed that BSV images with RNBs are rare, accounting for fewer than 5 % of the sampled images. To alleviate the impact of class imbalance problem on model training, 50,000 images are randomly selected from each city tier based on the training sampling set. These samples are labelled as positive type (i.e., image with RNB) or negative type (i.e., image without RNB). Subsequently, the same number of positive and negative samples are maintained. Certain objects,

such as tunnel inner walls, billboards, and guardrails, seem like RNBs in images, which intensifies the difficulty of deep learning, as shown in Fig. 4. Therefore, 500 images of each of these objects are added as confusing negative samples to the training samples. The ultimate training sample size is 14,484, including 6,492 positive and 7,992 negative samples. To generate the validation and test samples, 500 and 2,500 image samples from each city tier are chosen. There are 79 positive samples and 1,921 negative samples in the validation samples, while there are 350 positive samples and 9,650 negative

samples in the test samples. The details of sample collection results are shown in Table 1. The labelled BSV images are available at https://doi.org/10.11888/Others.tpdc.271914 (Chen, 2021).



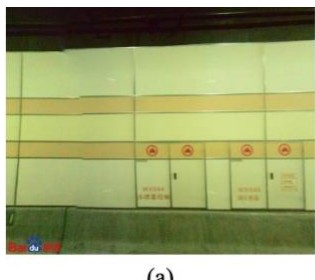 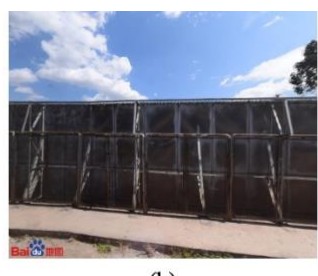 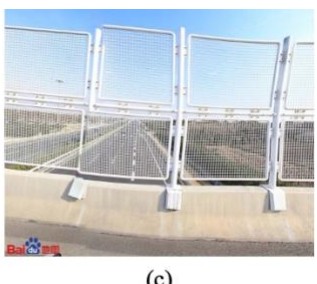

$$(a) \qquad\qquad (b) \qquad\qquad (c)$$

**Figure 4: Three confusing negative samples which look like RNBs, including (a) tunnel inner wall, (b) billboard, and (c) guardrail. (BSV images are from Baidu Maps)**

**Table 1: Details of sample collection results.**

| | Type | Tier 1 | Tier 2 | Tier 3 | Tier 4 | Confusing sample | Total |
|---|---|---|---|---|---|---|---|
| Training samples | Positive | 2,886 | 2,191 | 870 | 545 | / | 6,492 |
| | Negative | 2,886 | 2,191 | 870 | 545 | 1,500 | 7,992 |
| | Total | 5,772 | 4,382 | 1,740 | 1,090 | 1,500 | 14,484 |
| Validation samples | Positive | 40 | 18 | 18 | 3 | / | 79 |
| | Negative | 460 | 482 | 482 | 497 | / | 1,921 |
| | Total | 500 | 500 | 500 | 500 | / | 2,000 |
| Test samples | Positive | 129 | 115 | 77 | 29 | / | 350 |
| | Negative | 2,371 | 2,385 | 2,423 | 2,471 | / | 9,650 |
| | Total | 2,500 | 2,500 | 2,500 | 2,500 | / | 10,000 |

## 2.3 Modelling

### 2.3.1 Convolutional neural network incorporating image context information (IC-CNN)

RNBs are widely placed on the roadside in densely populated regions, such as residential, educational, and government institutions, as previously described in studies (Arenas, 2008; Wang et al., 2018; Zhang et al., 2022). Therefore, based on this prior geographic knowledge, an IC-CNN that leverages the context information contained in BSV images is developed, which aims at enhancing the RNB detection accuracy. Figure 5 illustrates the construction of IC-CNN, which adopts the ResNet architecture (He et al., 2016). In this workflow, prior geographic knowledge is incorporated into the neural network by means of transferring learning. Initially, 500 samples are randomly selected from positive and negative training samples in each tier. Three context labels are added depending on the context of these BSV images: building dominated, non-building dominated, and uncertain (unable to judge the background of the BSV image because it is obscured by objects). The context labels are interpreted by semantic segmentation models released by MIT Computer Vision team (Zhou et al., 2019a). Besides the sky and ground objects, images are judged to be building dominated if the building objects occupy the majority of the image; otherwise, they are evaluated to be non-building dominated. Additionally, the uncertain type is classified by visual



interpretation   of   whether   the   image   is   obscured.   These   labelled   images   are   available   at
https://doi.org/10.11888/Others.tpdc.271914 (Chen, 2021). Next, 4,000 samples with image context labels are used to train
the IC-CNN on a preliminary basis, where using hybrid loss to optimize parameters in IC-CNN for image context and RNB
identification, as formulated in Eq. (1). After the network has converged, the IC-CNN's classifier is replaced with a binary
classification, and all the training samples are supplemented to fine-tune and intensively train the network.

**Figure 5: The construction of convolutional neural network incorporating image background information. (BSV images are from Baidu Maps)**

$$\text{Hybrid loss} = \text{CE}\left(p_{image\ context}\right) + 2 \times \text{CE}\left(p_{noise\ barrier}\right), \tag{1}$$

$$\text{CE}(p) = -\sum p \cdot \log(p), \tag{2}$$





where $p_{image\ context}$ is the confidence of image context identification, $p_{noise\ barrier}$ is the confidence of RNB identification,
and CE(p) refers to the cross-entropy loss function (Hu et al., 2018).

### 2.3.2 Ensemble learning strategy

Owing to the high cost of labelling and the restricted quantity of trained samples, an ensemble learning strategy for
enhancing RNB detection accuracy is utilized in this study based on the "no free lunch" theory (Wolpert and Macready,
1997). In ensemble learning domain, the effective strategy to boost performance is to integrate the numerous high-variance
models together (Cao et al., 2020). Therefore, this study integrates four IC-CNNs, and their convolutional layers are chosen
from the ResNet family (He et al., 2016; Zagoruyko and Komodakis, 2016), including ResNet101, ResNet152, Wide
ResNet50, and Wide ResNet101. The integration of the four IC-CNNs with varying capacities for feature extraction can
make a significant contribution to achieve high detection accuracy.

### 2.3.3 Vectorization post-processing

After performing detection by an ensemble of IC-CNNs, the identified and scattered RNB locations are connected to create a
vectorized RNB dataset by a post-processing technique, which is based on the spatial neighbour relationship between
samples. Specifically, if adjacent sample images of the same road contain RNB objects, their locations will be connected.
Furthermore, the findings of Sainju and Jiang (2020) demonstrated that "near objects are more related" principle (Tobler,
1970, 2004) holds true when using street view imagery to detect objects at the urban scale. Therefore, in this study, given the
likelihood of RNB misidentification, if a sample image is flanked by images containing RNBs in the same way, it will be
considered as a positive type to minimize the impact of misidentification.

### 2.4 Evaluation methods

### 2.4.1 Metrics for detection accuracy

To evaluate the accuracy of RNB detection, four quantitative metrics in the deep learning classification task, including
overall accuracy (OA), recall, precision, and F1-score (Thomas et al., 2020) are analyzed. After detecting the RNBs in BSV
images, the number of false-negative (FN), true-negative (TN), true-positive (TP), and false-positive (FP) images is
calculated. True positive means the prediction and ground truth of images are both positive. Conversely, false negative
means the predictions are negative while the ground truths are positive. The four metrics are calculated based on the
following Eqs. (3)-(6) (Thomas et al., 2020):

$$OA = \frac{TP+TN}{TP+FP+TN+FN},\tag{3}$$

$$Precision = \frac{TP}{TP+FP},\tag{4}$$





$$Recall = \frac{\text{TP}}{\text{TP+FN}}, \tag{5}$$

$$F1 - score = \frac{2\cdot\text{Precision}\cdot\text{Recall}}{\text{Precision+Recall}}, \tag{6}$$

### 2.4.2 Metrics for completeness and positional accuracy

To quantitatively evaluate completeness and positional accuracy of generated RNBs, two quantitative metrics, including RMSE and IoU are adopted (Rezatofighi et al., 2019). For calculating these metrics, numerous roads are selected from various cities and are surveyed manually as ground truths based on BSV imagery. Based on the mileage deviation and overlap relationship between the generated and surveyed RNBs, RMSE and IoU are calculated following Eqs. (7) and (8), respectively:

$$RMSE = \sqrt{\frac{1}{m}\sum_{i=1}^{m}(l_i - \widehat{l_i})^2}, \tag{7}$$

where m is the number of selected roads, $l_i$ is the surveyed RNB mileage of the i[th] road, and $\widehat{l_i}$ is the generated RNB mileage of the i[th] road.

$$IoU = \frac{\text{L}_{intersection}}{\text{L}_{union}}, \tag{8}$$

where $\text{L}_{intersection}$ is intersection mileage of generated and surveyed RNB, and $\text{L}_{union}$ is union mileage of generated and 235   surveyed RNB.

### 2.5 Implementation configuration

  Several techniques to enhance the performance of the model throughout the training and inference stages are employed in this study. Data augmentation techniques such as random resized cropping and random horizontal flipping are utilized to increase data volume and decrease model bias error. Subsequently, the model parameters are optimized using the cosine 240   annealing learning rate scheduler (Bhattacharyya et al., 2021) and AdamW optimizer (Loshchilov and Hutter, 2017). Additionally, long training and inference resized tuning (Touvron et al., 2019) are employed to improve the model's performance. Finally, an ensemble of models identifies RNBs based on the voting mechanism.

### 3 Results

### 3.1 RNB mapping result

The final RNBs dataset are available at https://doi.org/10.11888/Others.tpdc.271914 (Chen, 2021). Details of the BSV image identification results are shown in Appendix A, and details of RNB mileage by city in China are shown in Appendix B, with the total RNB mileage of 2227 km. The spatial distribution of RNB mileage among cities is depicted in Fig. 6, where blank

areas indicate no RNBs or lack of BSV images. Figure 6 suggests that RNBs in eastern China are more densely distributed and have longer mileage. Furthermore, Tier 1 and Tier 2 contain major portion of the total RNB mileage. Because unique urban administration system in China mandates lower-tier cities to rigidly follow the "leadership" of higher-tier cities (Ma, 2005; Zhao et al., 2003), higher-tier cities are rapidly increasing in size and occupying considerable resources, while lower-tier cities are developing slowly (Au and Henderson, 2006; Lin, 2002). Therefore, to a certain extent, it shows that the statistics correlate with the development of Chinese cities, implying that higher-tier cities have a high probability of covering and updating BSV imagery or laying down RNBs.

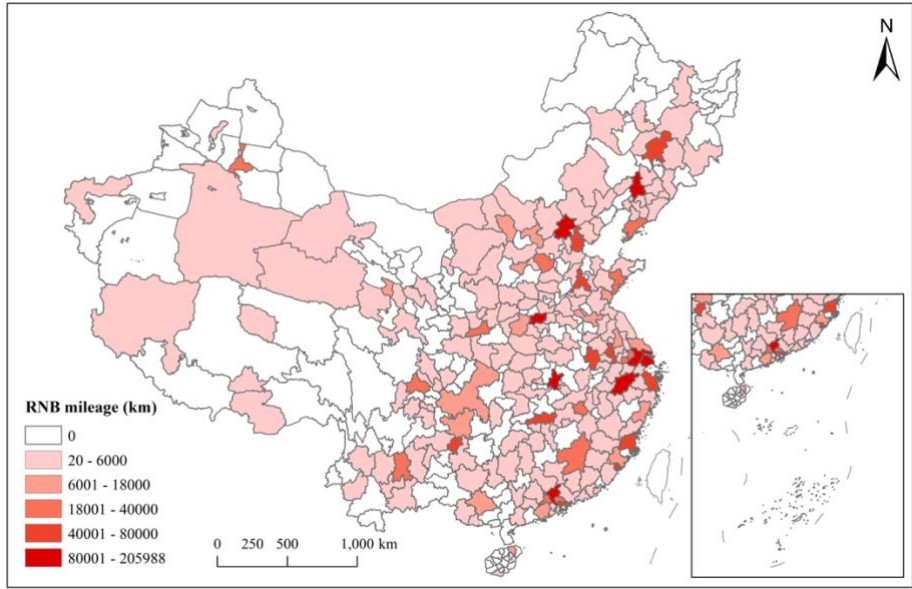

**Figure 6: Mileage zonal statistic in China. The blank areas indicate no RNBs or lack of BSV images.**

After analyzing the generated RNB dataset from a national scale, three cities with the highest RNB mileage in each tier are selected to analyze the citywide mapping results, as shown in Fig. 7. The figure shows that RNBs are generally clustered in the central areas of these cities. For example, the RNBs in Shanghai are mainly clustered on the third ring road, while those in Beijing are mainly clustered on the sixth ring road. As a result, when combined with the planed layout and actual mapping of RNB distribution, the data can partially reflect the rationality of urban infrastructure planning and layout.



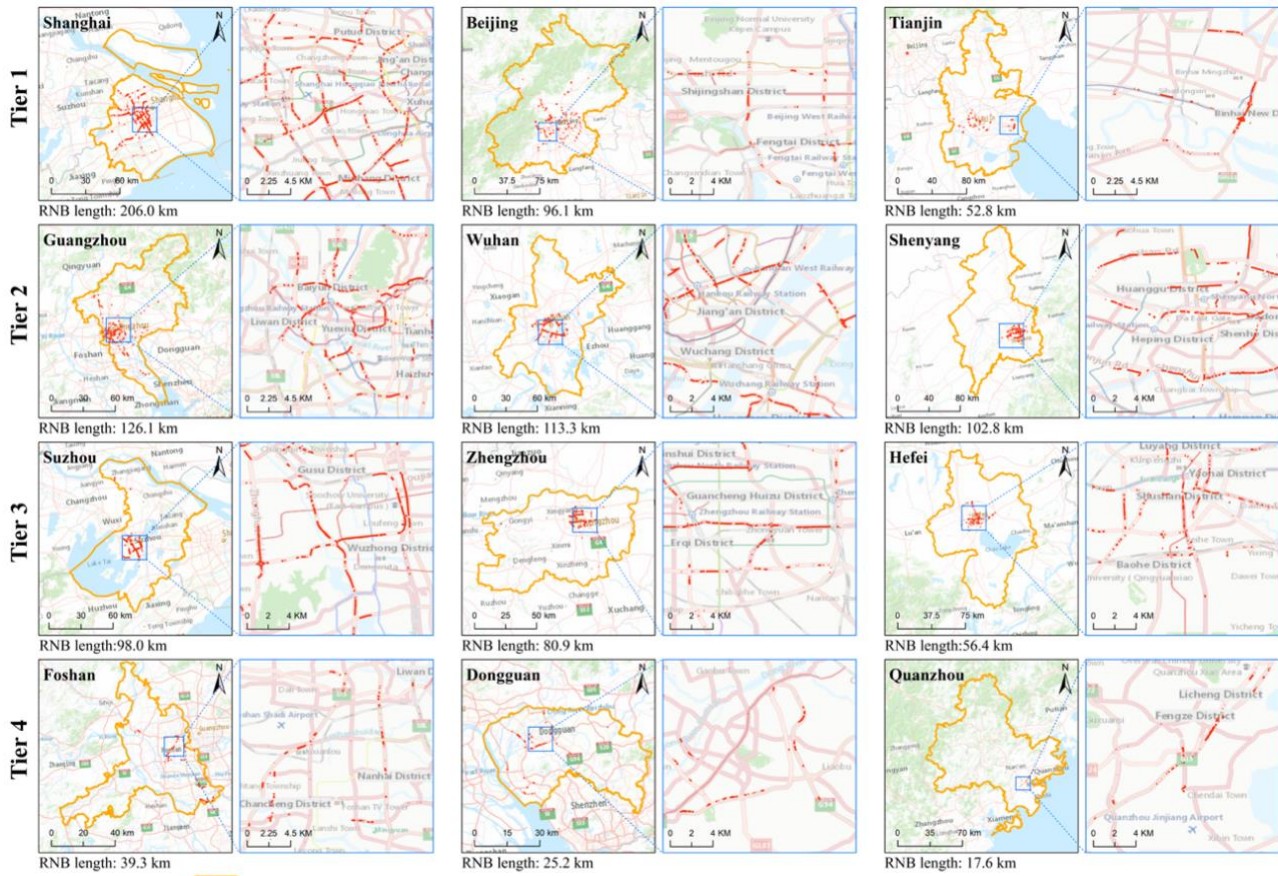

**Figure 7: Distribution of RNBs in several representative cities. (BSV images are from Baidu Maps and base maps are from ESRI)**

## 3.2 Evaluation and analysis

### 3.2.1 RNB detection accuracy

Table 2 summarizes the evaluation results of RNB identification at different city tiers based on test samples. The OA and the F1-score for the overall city tiers are 98.61 % and 81.44 %, respectively. However, the accuracy is greater for higher-tier cities than for lower-tier cities. This may be attributed to the fact that cities with lower tiers appear to have a more severe class imbalance problem for deep learning methods, which affects the training and generalization of the model. Therefore,

the results indicate that prior to using this dataset, an assessment of the influence of regional quality differences on specific applications is required.

**Table 2: Evaluation results of RNB identification in different city tiers. The evaluation results of every city tier are calculated using the test samples of the corresponding city tier, while the overall evaluation results are calculated using the entire test samples.**

| City tier | OA (%) | Recall (%) | Precision (%) | F1-score (%) |
|---|---|---|---|---|





| | | | | |
|---|---|---|---|---|
| Tier 1 | 98.12 | 88.37 | 78.08 | 82.91 |
| Tier 2 | 98.28 | 86.09 | 78.57 | 82.16 |
| Tier 3 | 98.68 | 87.01 | 74.44 | 80.24 |
| Tier 4 | 99.36 | 86.21 | 67.57 | 75.76 |
| Overall | 98.61 | 87.14 | 76.44 | 81.44 |

### 3.2.2 RNB completeness and positional accuracy

To evaluate the completeness and positional accuracy of the RNB dataset, approximately 254 km of roads are selected from different city tiers and manually surveyed using the BSV imagery. Appendix C summarizes the detailed quantitative differences between generated and surveyed RNBs in terms of mileage deviation and level of overlap. The overall RMSE for mileage deviation is 0.08 km and IoU for overlay level is 88.08 % ± 2.95 %. The results shows that the generated and surveyed RNBs are highly consistent in terms of mileage and distribution, demonstrating the high completeness and positional accuracy of the generated RNB dataset.

Moreover, as illustrated in Fig. 8, the visual comparison between surveyed and generated RNBs on various roads depicts that the generated and surveyed RNBs on the road are overall consistent in terms of mapping. However, several validated points demonstrated that the proposed deep learning approach incorrectly recognized small RNB objects in the images, such as validated points IV, II, and III on Beijing's Jingmen motorway, Zhengzhou's Longhai Road, and Wenzhous's Ouhai Avenue, respectively. Additionally, several objects that looked similar to RNBs, such as multi-windowed buildings, are misclassified as positive type; for example, point IV on Wenzhou's Ouhai Avenue. Despite these misclassifications, most of the validated points demonstrated a high accuracy of the RNB prediction and the high performance of the proposed framework, implying the reliability of the generated RNB dataset.





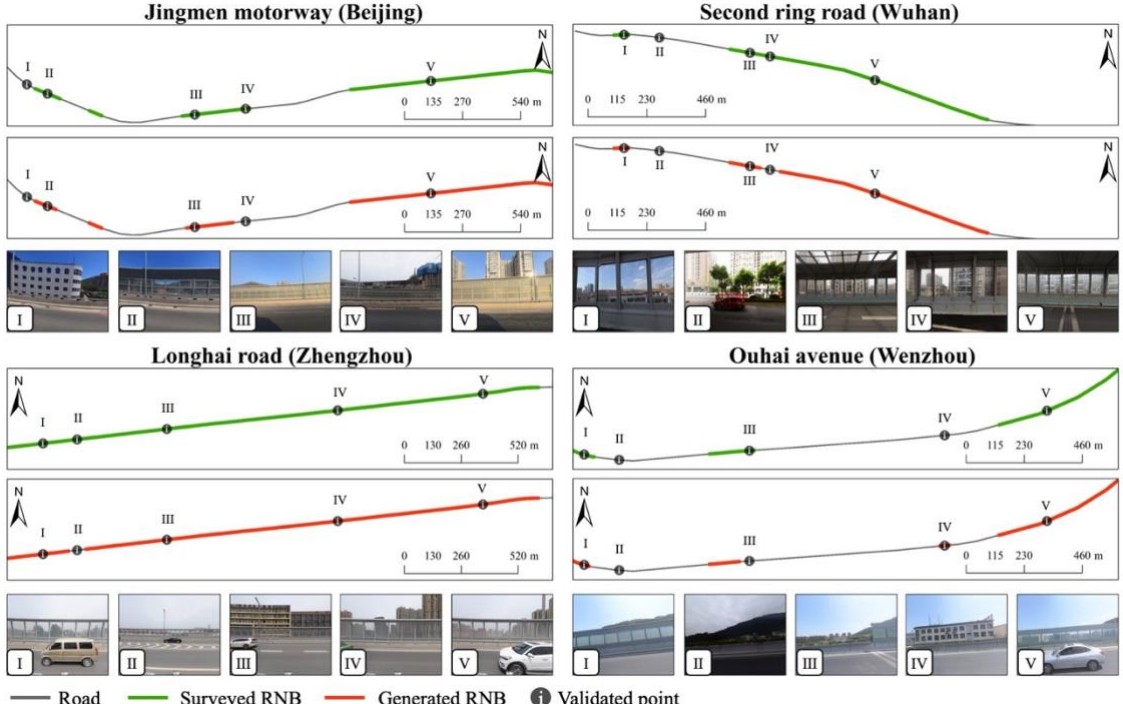


**Figure 8: RNB mapping result in city scale. (BSV images are from Baidu Maps)**

## 4 Discussion

### 4.1 Model capability

An ablation study is conducted to demonstrate the quality of the generated dataset and validate the effectiveness of
developed methods (Table 3). As shown in Table 4, the combination of proposed strategies achieves the highest performance.
The ablation results illustrate that the effectiveness of proposed strategies, including integrating image context information
into CNN, adding confusing negative samples, and ensemble learning strategy. Additionally, Figure 9 depicts the areas of
IC-CNNs' attention, revealing that IC-CNNs not only have a capacity for focusing on RNB objects in BSV images, but also
have a sense of their surrounds. The results suggest the reliability of the generated dataset and partially decipher the "black
box" of deep learning to explain the high performance of the developed methods. Notably, this study successfully achieves
incorporating some of the prior geographic knowledge into the deep learning method. RNB detection accuracy can be
increased further by combining more comprehensive knowledge of geographic scenes from BSV images into deep learning
network, such as various geographic elements and processes as well as the associated construction theory (Lü et al., 2018).

**Table 3: Ablation study design. The ablation study combines the four strategies used in this study to illustrate their effectiveness.**



| Ablation | Baseline (ResNet101) | Incorporate image context information | Add confusing negative samples | Ensemble learning strategy |
|---|---|---|---|---|
| I | ✓ | | | |
| II | ✓ | ✓ | | |
| III | ✓ | ✓ | ✓ | |
| IV | ✓ | ✓ | ✓ | ✓ |

**Table 4: Quantitative results of ablation. The ablation results show that the proposed methods have the highest RNB detection accuracy.**

| Ablation | OA | Recall | Precision | F1-score |
|---|---|---|---|---|
| I | 97.81 % (± 0.01 %) | 62.91 % (± 0.41 %) | **74.14 % (± 0.16 %)** | 64.62 % (± 0.25 %) |
| II | 97.50 % (± 0.03 %) | 86.00 % (± 0.09 %) | 63.67 % (± 0.25 %) | 72.05 % (± 0.15 %) |
| III | 98.02 % (± 0.01 %) | 81.71 % (± 0.07 %) | 68.82 % (± 0.13 %) | 74.41 % (± 0.07 %) |
| **IV** | **98.32 % (± 0.00 %)** | **85.60 % (± 0.08 %)** | 71.87 % (± 0.04 %) | **78.09 % (± 0.05 %)** |




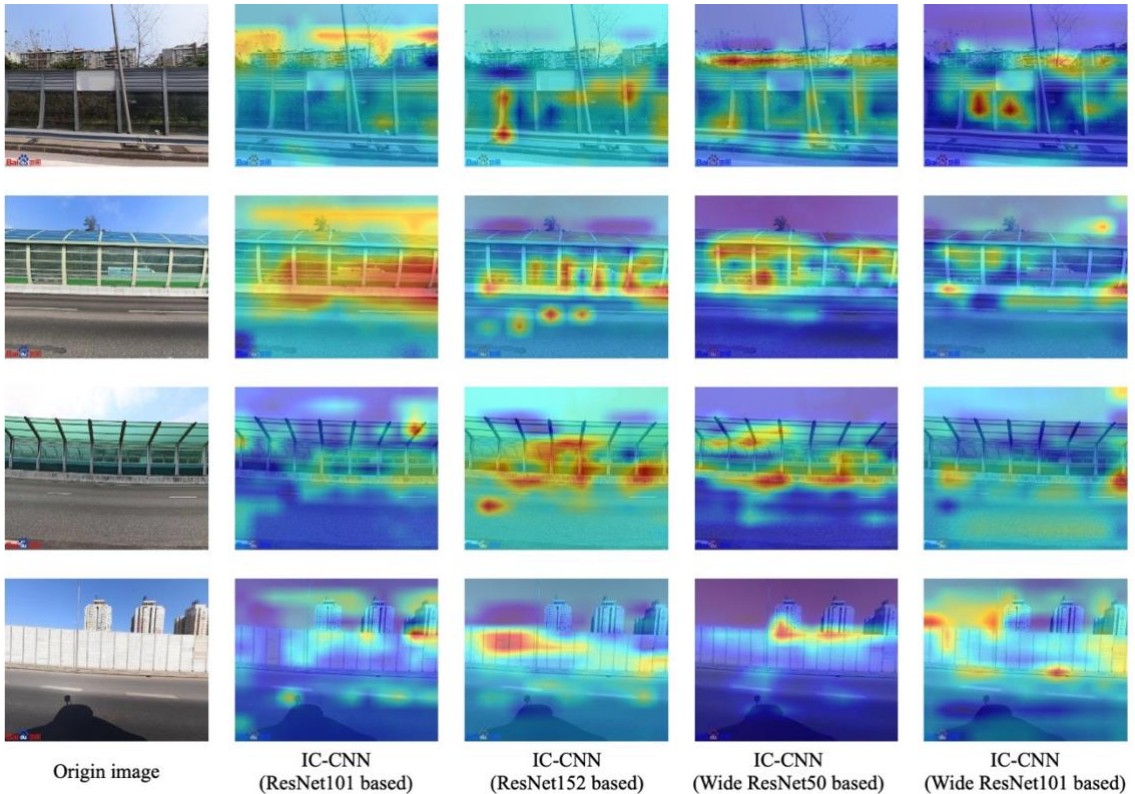

**Figure 9: Attention maps of IC-CNNs on BSV images with RNB. The hotspots indicate the area where the attention of IC-CNN is focused. (BSV images are from Baidu Maps)**

### 4.2 Limitations and future work

This study has several limitations in the process of dataset generation, which can be grouped into three categories, namely data source, ground scenario, and modelling:

(1) Due to economic status, topographical conditions, or government policies, not all Chinese cities are covered by BSV imagery, with data not available for 18 cities (Deng et al., 2021; Du et al., 2020). In addition, challenges owing to overexposure or obstruction of the sensors by vehicles hinder capturing a complete street scene. As a result, the natural characteristics of the data source can have certain impacts on the accuracy of the RNB dataset.

(2) The road/traffic environment is often complex. Concretely, BSV sensors can detect RNBs on distant highways or other lanes, and it may result in some mistakes during RNB detection and mapping. However, the likelihood of this occurring is small (about 4 % of RNB samples) by sampling investigation.

(3) This study implicitly presupposes that BSV images are independent and identically distributed. As shown in Fig. 8, the developed GeoAI framework can achieve high performance in continuous RNB mapping. However, spatial autocorrelation effect in BSV images is overlooked, as BSV images taken along the same road network path frequently resemble adjacent one (Sainju and Jiang, 2020).





In the future, to address the data shortage issue, more data sources, such as Google Maps and Tencent Maps, will be used.
Additionally, approaches for photogrammetry and image scene understanding techniques will be developed to tackle the complex ground scenario. Finally, end-to-end deep learning algorithms will be constantly enhanced by the addition of more powerful units and structures to account for spatial autocorrelation in street view imagery.

## 5 Code availability

The codes of deep learning approaches in this study are available at https://doi.org/10.11888/Others.tpdc.271914 (Chen,
2021) and https://github.com/ChanceQZ/NoiseBarrierIdentification. Python3 packages such as PyTorch, NumPy, and OpenCV are used to develop the code. The vectorization post-processing procedure is performed in the ArcGIS Pro platform.

## 6 Data availability

The road data comes from OSM (https://www.openstreetmap.org/), a collaborative project dedicated to providing many
types of freely editable geographic data for the world. City boundaries can be obtained from http://bzdt.ch.mnr.gov.cn/. In addition, BSV images can be downloaded by using BSV API (https://api.map.baidu.com/panorama/v2?key=parameters). Finally, the generated RNB dataset, labelled BSV image benchmark, and RNB detection results are available to the public at https://doi.org/10.11888/Others.tpdc.271914 (Chen, 2021). The mileage of RNB is calculated in Albers equal-area conical projection.

## 7 Conclusion

This study presents the first nationwide RNB dataset in China using BSV imagery based on a GeoAI framework as well as the labelled BSV image benchmark. In this study, based on prior geographic knowledge in BSV imagery, RNB samples are identified based on deep learning approaches. Subsequently, the vectorized RNB dataset is constructed using the post-processing procedure. Finally, the created RNB dataset is evaluated from two perspectives, i.e., the detection accuracy as
well as the completeness and positional accuracy. The four quantitative metrics, OA, recall, precision, and F1-score, analyzed are all high, showing high accuracy of the model in RNB detection. Additionally, the level of mileage deviation and overlay between the generated and surveyed RNBs are determined via a manual survey of around 254 km of roads in various cities. The RMSE for mileage deviation and the IoU for overlay level revealed that the created and surveyed RNBs are consistent. The results indicate that the created RNB dataset can serve as a reliable dataset for local governments and
urban research institutions in terms of data support and decision-making., with the support of the RNB dataset, the improved energy conversion estimation at a large scale can enable more precise modeling and analysis. Besides, the dataset can be





used to assist in urban planning and regional economic research. Furthermore, the labelled BSV image benchmark aids in the development and training of deep learning models for additional urban studies.




## 355 Appendix A

**Table A1: Details of the BSV image identification results.**

| City tier | Negative (BSV image count) | Positive (BSV image count) | Total (BSV image count) |
|---|---|---|---|
| Tier 1 | 636,566 | 48,159 | 684,725 |
| Tier 2 | 1,594,057 | 137,686 | 1,731,743 |
| Tier 3 | 1,308,692 | 83,264 | 1,391,956 |
| Tier 4 | 1,746,742 | 34,525 | 1,781,267 |
| Overall | 5,286,057 | 303,634 | 5,589,691 |

**Table A2: Identification confusion matrix based on test samples.**

| Tier 1 | | Predicted class | |
|---|---|---|---|
| | | Negative | Positive |
| True class | Negative | 2,339 | 32 |
| | Positive | 15 | 114 |

| Tier 2 | | Predicted class | |
|---|---|---|---|
| | | Negative | Positive |
| True class | Negative | 2,358 | 27 |
| | Positive | 16 | 99 |

| Tier 3 | | Predicted class | |
|---|---|---|---|
| | | Negative | Positive |
| True class | Negative | 2,400 | 23 |
| | Positive | 10 | 67 |

| Tier 4 | | Predicted class | |
|---|---|---|---|
| | | Negative | Positive |
| True class | Negative | 2,459 | 12 |
| | Positive | 4 | 25 |

| Overall | | Predicted class | |
|---|---|---|---|
| | | Negative | Positive |
| True class | Negative | 9,556 | 94 |
| | Positive | 45 | 305 |



## Appendix B

The total RNB mileage in China is 2226.85 km. The RNB mileage in different city tiers are 369.42 km, 941.72 km, 605.21 km, and 310.49 km, respectively.

**Table B1: Details of RNB mileage by city in China. The RNB mileages of some cities are 0 km, indicating that they lack RNBs or BSV images, or that the BSV images are out of date.**

| Tier 1 | | Tier 4 | | | | | |
|---|---|---|---|---|---|---|---|
| **City** | **Mileage (km)** | **City** | **Mileage (km)** | **City** | **Mileage (km)** | **City** | **Mileage (km)** |
| Shanghai | 205.99 | Foshan | 39.27 | Xinxiang | 1.05 | Suqian | 0.12 |
| Beijing | 96.13 | Dongguan | 25.22 | Huangshi | 1.04 | Leshan | 0.12 |
| Tianjin | 52.75 | Ganzhou | 20.91 | Hainan | 0.99 | Wuzhou | 0.11 |
| Chongqing | 13.33 | Nantong | 17.70 | Taizhou (Zhe) | 0.95 | Shaoyang | 0.10 |
| Macao | 1.23 | Quanzhou | 17.62 | Hanzhong | 0.95 | Qionghai | 0.10 |
| **Tier 2** | | Wenzhou | 11.03 | Anyang | 0.79 | Chongzuo | 0.10 |
| **City** | **Mileage (km)** | Yangzhou | 10.80 | Jiaxing | 0.78 | Shangqiu | 0.09 |
| Guangzhou | 126.14 | Jiangmen | 9.39 | Jiayuguan | 0.78 | Jingdezhen | 0.09 |
| Wuhan | 113.33 | Zunyi | 9.29 | Jiujiang | 0.74 | Xiangxi | 0.08 |
| Shenyang | 102.77 | Rizhao | 6.31 | Xianyang | 0.71 | Xuchang | 0.08 |
| Hangzhou | 93.62 | Deyang | 5.61 | Liaoyang | 0.70 | Xuancheng | 0.08 |
| Nanjing | 73.24 | Linyi | 5.59 | Jincheng | 0.67 | Huangshan | 0.08 |
| Jinan | 72.60 | Kaifeng | 5.56 | Panjin | 0.66 | Xiangtan | 0.08 |
| Ningbo | 72.01 | Yichang | 5.38 | Pingdingshan | 0.65 | Bijie | 0.08 |
| Shenzhen | 54.52 | Chifeng | 4.74 | Longyan | 0.64 | Pingxiang | 0.08 |
| Xiamen | 50.25 | Maanshan | 4.06 | Bayinguoleng | 0.62 | Changzhi | 0.07 |
| Changchun | 44.61 | Zhuhai | 3.87 | Qingyuan | 0.62 | Liupanshui | 0.07 |
| Dalian | 37.54 | Xingtai | 3.76 | Nanping | 0.58 | Yichun (Hei) | 0.07 |
| Qingdao | 36.71 | Zhenjiang | 3.59 | Nanchong | 0.56 | Zhangzhou | 0.06 |
| Chengdu | 33.04 | Baoji | 3.55 | Liuzhou | 0.54 | Meizhou | 0.06 |
| Xian | 26.14 | Chanzhou | 3.48 | Zhangjiakou | 0.51 | Ezhou | 0.06 |
| Harbin | 5.22 | Shantou | 3.09 | Sanming | 0.50 | Hinggan | 0.06 |
| **Tier 3** | | Weifang | 2.80 | Yaan | 0.49 | Fuxin | 0.05 |
| **City** | **Mileage (km)** | Huizhou | 2.78 | Zhuzhou | 0.47 | Haidong | 0.05 |
| Suzhou (Su) | 97.96 | Weinan | 2.76 | Jiuquan | 0.35 | Tongchuan | 0.05 |
| Zhengzhou | 80.92 | Zhaoqing | 2.67 | Cangzhou | 0.34 | Yichun (Gan) | 0.05 |



| | | | | | | | |
|---|---|---|---|---|---|---|---|
| Hefei | 56.38 | Hengyang | 2.66 | Nanyang | 0.33 | Yingkou | 0.05 |
| Guiyang | 48.71 | Jinghua | 2.65 | Heyuan | 0.33 | Honghe | 0.04 |
| Changsha | 47.88 | Baoding | 2.57 | Tieling | 0.28 | Gannan | 0.04 |
| Fuzhou (Min) | 46.30 | Huzhou | 2.55 | Qinhuangdao | 0.28 | Zigong | 0.04 |
| Nanchang | 36.03 | Xiangyang | 2.51 | Tianmen | 0.26 | Shaoguan | 0.04 |
| Urumqi | 30.09 | Haixi | 2.43 | Kizilsukirk | 0.26 | Bayannaoer | 0.04 |
| Kunming | 23.57 | Taian | 2.35 | Xinzhou | 0.25 | Qujing | 0.04 |
| Wuxi | 22.95 | Ordos | 2.17 | Changde | 0.25 | Chuxiong | 0.04 |
| Shijiazhuang | 21.32 | Sanya | 2.16 | Tonghua | 0.25 | Suining | 0.04 |
| Naning | 15.84 | Mianyang | 2.08 | Fuzhou (Gan) | 0.24 | Dezhou | 0.03 |
| Taiyuan | 13.01 | Wuhu | 2.05 | Guilin | 0.24 | Zhumadian | 0.02 |
| Xuzhou | 12.47 | Shangrao | 2.01 | Yunfu | 0.23 | Chuzhou | 0.02 |
| Xining | 8.93 | Lianyungang | 1.97 | Dandong | 0.22 | Yuncheng | 0.02 |
| Hohhot | 8.78 | Taizhou (Su) | 1.93 | Jingzhou | 0.21 | Fuyang | 0.02 |
| Haikou | 7.74 | Shaoxing | 1.89 | Yanan | 0.21 | Chaoyang | 0.02 |
| Datong | 7.45 | Dali | 1.74 | Xinyu | 0.20 | Lincang | 0.02 |
| Luoyang | 7.42 | Chengde | 1.55 | Ningde | 0.20 | Ankang | 0.02 |
| Lanzhou | 4.09 | Wuhai | 1.53 | Qiannan | 0.20 | Shanwei | 0.02 |
| Zibo | 1.67 | Zhongshan | 1.47 | Yangquan | 0.19 | Fangchenggang | 0.02 |
| Anshan | 1.25 | Yuxi | 1.45 | Yulin (Qin) | 0.19 | Yongzhou | 0.02 |
| Tangshan | 1.19 | Yanbian | 1.43 | Yibin | 0.19 | Jieyang | 0.02 |
| Handan | 1.15 | Daqing | 1.34 | Langfang | 0.17 | Maoming | 0.02 |
| Yinchuan | 1.05 | Shiyan | 1.26 | Ulanqab | 0.17 | Hechi | 0.02 |
| Benxi | 0.43 | Yantai | 1.25 | Huaian | 0.17 | Shannan | 0.02 |
| Jilin | 0.33 | Anqing | 1.23 | Ali | 0.16 | Bengbu | 0.02 |
| Baotou | 0.18 | Yancheng | 1.21 | Jining | 0.16 | Quzhou | 0.02 |
| Lhasa | 0.06 | Songyuan | 1.21 | Sanmenxia | 0.15 | Jingmen | 0.02 |
| Fushun | 0.06 | Dongying | 1.11 | Baiyin | 0.13 | Xinyang | 0.02 |
| | | Heze | 1.08 | Karamay | 0.12 | Linfen | 0.02 |

The RNB mileages of other cities are 0 km.




## Appendix C

**Table C1: Quantitative comparison with the generated and surveyed RNBs in different roads in different city tiers. The 4 km - 7.5 km of roads with RNBs are selected as surveyed objects. The total road mileage is around 254 km.**

| Tier | City | Road name | Road mileage (km) | Surveyed RNB mileage (km) | Generated RNB mileage (km) | IoU (%) |
|---|---|---|---|---|---|---|
| 1 | Beijing | Guangqu motorway | 6.37 | 1.81 | 1.29 | 71.52 |
| | | Beijing-Urumqi motorway | 4.13 | 3.07 | 2.95 | 96.06 |
| | | Jingmen motorway | 5.23 | 1.58 | 1.46 | 92.41 |
| | Chongqing | Tushan road | 5.58 | 0.77 | 0.43 | 56.19 |
| | | Jichang road | 5.24 | 2.16 | 1.56 | 71.24 |
| | | Inner ring motorway | 4.63 | 0.39 | 0.34 | 89.41 |
| | Shanghai | Shanghai-Kunming motorway | 6.17 | 4.19 | 4.19 | 100.00 |
| | | Shanghai-Jinshan motorway | 6.55 | 5.50 | 5.34 | 97.07 |
| | | Humin elevated road | 6.43 | 3.10 | 3.10 | 92.69 |
| | Tianjin | Hongqi south road | 4.97 | 0.91 | 0.91 | 95.05 |
| | | Kunlun road | 5.03 | 2.01 | 1.89 | 93.80 |
| | | Ninghe-Jinghai motorway | 6.23 | 2.60 | 2.07 | 78.22 |
| 2 | Chengdu | No.2 Elevated ring road | 4.80 | 0.93 | 0.91 | 83.80 |
| | | Chengbei motorway | 4.54 | 2.32 | 2.32 | 100.00 |
| | | Cheng-Yu Area ring motorway | 4.65 | 3.14 | 2.15 | 68.48 |
| | Guangzhou | City ring motorway | 5.10 | 1.83 | 1.83 | 100.00 |
| | | Huanan motorway | 4.29 | 1.22 | 1.22 | 100.00 |
| | | Liede avenue | 4.85 | 0.89 | 0.98 | 91.20 |
| | Nanjing | Airport motorway | 5.95 | 1.43 | 0.91 | 63.35 |
| | | Shanghai-Chengdu motorway | 5.28 | 1.48 | 1.16 | 78.52 |
| | | Jiangbei avenue | 5.51 | 1.89 | 2.15 | 86.20 |
| | Wuhan | Longyang avenue | 4.99 | 0.61 | 0.74 | 77.76 |
| | | Second ring road | 5.23 | 2.31 | 2.31 | 93.35 |
| | | Baishazhou elevated road | 7.08 | 2.76 | 2.74 | 97.64 |
| 3 | Fuzhou | Airport motorway | 5.82 | 1.36 | 1.16 | 85.53 |
| | | East No.3 ring road | 4.60 | 1.38 | 1.34 | 96.43 |
| | | North No.3 ring road | 4.61 | 2.04 | 1.84 | 90.19 |
| | Hefei | Tongling road | 6.66 | 2.44 | 2.35 | 83.76 |
| | | North South No.1 elevated road | 6.35 | 2.66 | 2.23 | 83.98 |





| | | Co-operative south road | 4.51 | 2.04 | 1.99 | 97.66 |
|---|---|---|---|---|---|---|
| | Suzhou | Youxin motorway | 6.27 | 2.75 | 2.73 | 99.21 |
| | | South ring motorway | 7.15 | 4.99 | 4.85 | 96.42 |
| | | Central west road | 5.79 | 3.34 | 2.98 | 89.36 |
| | Zhengzhou | Longhai east road | 4.51 | 3.19 | 2.88 | 85.44 |
| | | Longhai road | 4.55 | 2.62 | 2.54 | 96.99 |
| | | East No.3 ring road | 4.80 | 1.01 | 1.20 | 73.72 |
| 4 | Dongguan | South ring road | 4.88 | 1.04 | 0.94 | 90.29 |
| | | Shenyang-Haikou motorway | 4.70 | 1.86 | 1.91 | 95.68 |
| | | Huancheng Road | 4.72 | 0.91 | 0.87 | 95.30 |
| | Nantong | Changjiang middle road | 4.46 | 1.86 | 1.80 | 96.57 |
| | | Hongjiang elevated road | 5.07 | 0.85 | 0.80 | 94.40 |
| | | Binjiang bridge | 6.14 | 1.70 | 1.98 | 79.75 |
| | Quanzhou | Shenyang-Haikou motorway | 4.21 | 2.45 | 1.79 | 73.16 |
| | | Huacheng south road | 5.73 | 1.08 | 1.10 | 94.21 |
| | | Airport motorway | 4.73 | 0.68 | 0.61 | 90.32 |
| | Wenzhou | Ouhai avenue | 4.74 | 2.31 | 2.08 | 87.12 |
| | | National highway 104 | 5.46 | 0.31 | 0.35 | 88.31 |
| | | Wenzhou bridge | 5.16 | 0.66 | 0.60 | 90.00 |




**Author contribution**

ZQ developed the framework, performed experiments, and wrote the original draft. MC conceptualized and supervised the project, as well as contributed with the design of the work and the critical revision of the article. YYang collected and processed data source, as well as published dataset. KZ collected and processed data source, as well as published dataset. TZ contributed with the design of the work and the critical revision of the article. FZ contributed with the design of the work and the critical revision of the article. RZ contributed with the design of the work and the critical revision of the article. ZXZ contributed with the design of the work and the critical revision of the article. ZS aided in data preparation. PLM aided in data collection and visualization. GNL contributed with technical review. YYu contributed with technical review.

**Competing interests**

The authors declare that they have no conflict of interest.

**Acknowledgements**

We appreciate the detailed suggestions and comments from the anonymous reviewers. We express heartfelt thanks to the other members of the Smart City Sensing and Simulation lab as well as OpenGMS lab, who undertook data collection and annotation work. The data of this work is licensed and hosted by National Tibetan Plateau Data Center.

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
