# Peer review of "Vectorized dataset of roadside noise barriers in China using street view imagery"

_Earth System Science Data, 2022_

## Community Comment (CC3)

**Responses to RC1:**

*This paper creates a vectorized RNB dataset, which is an impressive work of good quality. This dataset will be beneficial for further studies as it is not always easy to create and find such data. Also, the street view image benchmark dataset is provided, which can be used as a training dataset for further work. There are some comments for the authors to consider:*

**Response:** We appreciate the favorable feedback and insightful comments from the reviewer. We believe the revised manuscript will address all the comments. Our responses to each comment are presented as follows.

*1. Do the authors see any possibility to extent the application of this approach to other regions outside of China in the future?*

**Response:** As far as we know, Google Street View covers a large range of places outside of China, in addition to having significant volunteer geographic information data. We will continue to collect such publically available data in the future, and if the possibility arises, we will create a larger vector road sound barrier dataset.

*2. I hope this dataset can be updated regularly to follow the frequency of Baidu Maps adaptation, and it is very beneficial, although the workload is enormous.*

**Response:** Thank you for bringing up this idea. As you might expect, given the frequency with which Baidu Street View images are updated, we gathered China-wide street view images again this year and updated the roadside noise barrier dataset, which we will update in the appropriate place in the article. In the future, we will update this dataset regularly to keep it up-to-date.

*3. Line 247-248: As said in this article, "where blank areas indicate no RNBs or lack of BSV images", from my perspective, it will be exciting and crucial to know exact information from blank areas, such as which cities lack RNBs or BSV images.*

**Response:** Thank you for suggestion. We will supplement this information in the revised manuscript.

---

## Author Response (AR1)

The authors are appreciative of the in-depth evaluation of our paper, which we believe will significantly enhance the manuscript. The manuscript has been revised, and it will be proofread by other peers before submission. In the marked-up version of the revised manuscript, the revisions for Referee #1 are highlighted in grey, while those for Referee #2 are highlighted in yellow. In addition, if the figure is revised, in the marked-up version of the revised manuscript, the top is new and the bottom is original (following MicroSoft Word's rules).

**Responses to RC1:**

*This paper creates a vectorized RNB dataset, which is an impressive work of good quality. This dataset will be beneficial for further studies as it is not always easy to create and find such data. Also, the street view image benchmark dataset is provided, which can be used as a training dataset for further work. There are some comments for the authors to consider:*

**Response:** We appreciate the favorable feedback and insightful comments from the reviewer. We believe the revised manuscript will address all the comments. Our responses to each comment are presented as follows.

*1. Do the authors see any possibility to extent the application of this approach to other regions outside of China in the future?*

**Response:** As far as we know, Google Street View covers a large range of places outside of China, in addition to having significant volunteer geographic information data. We will continue to collect such publically available data in the future, and if the possibility arises, we will create a larger vector road sound barrier dataset.

*2. I hope this dataset can be updated regularly to follow the frequency of Baidu Maps adaptation, and it is very beneficial, although the workload is enormous.*

**Response:** Thank you for bringing up this idea. As you might expect, given the frequency with which Baidu Street View images are updated, we gathered China-wide street view images again this year and updated the roadside noise barrier dataset, which we will update in the appropriate place in the article. In the future, we will update this dataset regularly to keep it up-to-date.

*3. Line 247-248: As said in this article, "where blank areas indicate no RNBs or lack of BSV images", from my perspective, it will be exciting and crucial to know exact information from blank areas, such as which cities lack RNBs or BSV images.*

**Response:** Thank you for suggestion. We will supplement this information in the revised manuscript.

**Responses to RC2:**

*The paper presented a nationwide dataset of roadside noise barriers (RNB) for China, which was mapped from street view photos using machine learning algorithms. According to the manuscript, the dataset was with high spatial accuracy and has potentials for urban studies. While the methodology and algorithm assessments seem reasonable to me, the paper is more like a technical report instead of an introduction of a useful dataset. My specific comments are:*

**Response:** We appreciate the reviewer's constructive ideas and remarks. We meticulously address each comment point-by-point, and corresponding contexts will be incorporated into the revised manuscript. We feel that our manuscript will be significantly enhanced with the assistance of your insightful comments and recommendations.

*1. Why the dataset is important? Are there any specific reasons why you created the dataset? Instead of some vague statements like "useful to a variety of urban studies", it would be more convincing to list some specific applications the dataset would have in China.*

**Response:** Roadside noise barriers (RNBs) are important urban infrastructures to develop a liveable city. As described in the Introduction section, RNBs have important uses in many ways, such as alleviating noise impact in communities (Abdulkareem et al., 2021; Ning et al., 2010), increasing the utility of new energy sources (Gu et al., 2012; Zhong et al., 2021), and improving roadside air quality (Huang et al., 2021; Zhao et al., 2021). With large-scale RNB datasets with detailed geospatial information, there are bottom-up benefits, such as managing and maintaining such infrastructure for municipalities (Sainju and Jiang, 2020), simulating and generating 3D models of dynamic cities (Wang and Wang, 2021; Zhao et al., 2017), and examining the sustainability of urban layouts (Song et al., 2021; Song and Wu, 2021).

Thank you for your suggestion. We are aware of the insufficient description of the importance and potential uses of RNBs in the Abstract and Conclusion sections, and we have added content to the revised manuscript to describe both in order to emphasize the significance of constructing this dataset.

**Reference:**

Abdulkareem, M., Havukainen, J., Nuortila-Jokinen, J., and Horttanainen, M.: Life cycle assessment of a low-height noise barrier for railway traffic noise, Journal of Cleaner Production, 323, 129169, https://doi.org/10.1016/j.jclepro.2021.129169, 2021.

Gu, M., Liu, Y., Yang, J., Peng, L., Zhao, C., Yang, Z., Yang, J., Fang, W., Fang, J., and Zhao, Z.: Estimation of environmental effect of PVNB installed along a metro line in China, Renewable energy, 45, 237-244, https://doi.org/10.1016/j.renene.2012.02.021, 2012.

Huang, Y., Lei, C., Liu, C. H., Perez, P., Forehead, H., Kong, S., and Zhou, J. L.: A review of strategies for mitigating roadside air pollution in urban street canyons, Environ Pollut, 280, 116971, https://doi.org/10.1016/j.envpol.2021.116971, 2021.

Ning, Z., Hudda, N., Daher, N., Kam, W., Herner, J., Kozawa, K., Mara, S., and Sioutas, C.: Impact of roadside noise barriers on particle size distributions and pollutants concentrations near freeways, Atmospheric Environment, 44, 3118-3127, https://doi.org/10.1016/j.atmosenv.2010.05.033, 2010.

Sainju, A. M. and Jiang, Z.: Mapping Road Safety Features from Streetview Imagery, ACM/IMS Transactions on Data Science, 1, 1-20, https://doi.org/10.1145/3362069, 2020.

Song, Y., Thatcher, D., Li, Q., McHugh, T., and Wu, P.: Developing sustainable road infrastructure performance indicators using a model-driven fuzzy spatial multi-criteria decision making method, Renewable and Sustainable Energy Reviews, 138, 110538, https://doi.org/10.1016/j.rser.2020.110538, 2021.

Song, Y. and Wu, P.: Earth Observation for Sustainable Infrastructure: A Review, Remote Sensing, 13, 1528, https://doi.org/10.3390/rs13081528, 2021.

Wang, S. and Wang, X.: Modeling and analysis of highway emission dispersion due to noise barrier and automobile wake effects, Atmospheric Pollution Research, 12, 67-75, https://doi.org/10.1016/j.apr.2020.08.013, 2021.

Zhao, W.-J., Liu, E.-X., Poh, H. J., Wang, B., Gao, S.-P., Png, C. E., Li, K. W., and Chong, S. H.: 3D traffic noise mapping using unstructured surface mesh representation of buildings and roads, Applied Acoustics, 127, 297-304, https://doi.org/10.1016/j.apacoust.2017.06.025, 2017.

Zhao, Y., Li, H., Kubilay, A., and Carmeliet, J.: Buoyancy effects on the flows around flat and steep street canyons in simplified urban settings subject to a neutral approaching boundary layer: Wind tunnel PIV measurements, Science of the Total Environment, 797, 149067, https://doi.org/10.1016/j.scitotenv.2021.149067, 2021.

Zhong, T., Zhang, K., Chen, M., Wang, Y., Zhu, R., Zhang, Z., Zhou, Z., Qian, Z., Lv, G., and Yan, J.: Assessment of solar photovoltaic potentials on urban noise barriers using street-view imagery, Renewable Energy, 168, 181-194, https://doi.org/10.1016/j.renene.2020.12.044, 2021.

*2. As mentioned, the current version of this manuscript is more like a technical paper, without comprehensive assessment of the dataset itself. Users would like to know more details about reliability and limitations of the dataset, such as spatial variations of mapping accuracy at the city scale, limitations in cities where limited street view photos are available, and the timing of detected RNB across cities. Given that the input street view photos were collected from 2014 to 2020 with uneven spatial distribution, knowing whether mapping accuracy was impacted by data completeness is particularly important. Without such information, it is hard to know whether the total 2227 km of RNB is reliable or not, as well as for each province.*

**Response:** We appreciate your suggestion. The technical sections of this manuscript (e.g., methodology explanation, technical result analysis, and model capability analysis) are essential for demonstrating the quality of our dataset. In order to conduct a full evaluation of the dataset, we've taken your advice and included analysis of spatial variation in mapping accuracy to the Results section, uncertainty analysis to the Discussion section, and timing information to the dataset itself and the Data Availability section. In addition, we update the figures that illustrate the specifics of the generated dataset.

*3. Technically, readers and users would like to know what types of RNB were mapped in the dataset and how the authors visually interpreted training, validation, and testing photos. It would be clearer to have some examples of RNB and detail logics used to manually label samples.*

**Response:** Thank you for suggestion. We have added descriptions and figures of distinct RNB types in revised manuscript. Moreover, we have further organized the descriptions of image labeling and added flowcharts to assist the reader in understanding our logics of labeling.

*4. It seems the deep learning architecture used in this study was adopted from others. I do not criticize the algorithm (although I do not think it's novel), but more specific design related to RNB detection should be considered and evaluated. Table 3 and 4 do not support that incorporating context information contributed to higher mapping accuracy.*

**Response:** In this paper, we develop a geospatial artificial intelligence framework based on the mature ResNet101 and Wide ResNet101 network models considering the street view data content and RNB characteristics, i.e., incorporating image context information, confusing negative samples, and ensemble learning strategies. Prior to this, we also tried more complex model baselines such as object detection algorithms, but the recognition results were not as good as our proposed approach.

Although Tables 3 and 4 show that the OA (overall accuracy) is not improved by incorporating the context information of BSV images, the integrated metric F1-score is improved significantly. In fact, F1-score is more representative in the task of positive and negative sample imbalance. We have supplemented the characteristics and importance of F1-score in the Data and Methods section.